# Toxicity of Tetracycline and Metronidazole in *Chlorella pyrenoidosa*

**DOI:** 10.3390/ijerph20043623

**Published:** 2023-02-17

**Authors:** Junrong Li, Yingjun Wang, Ziqi Fan, Panyang Tang, Mengting Wu, Hong Xiao, Zhenxing Zeng

**Affiliations:** 1Department of Environmental Engineering, College of Environment, Sichuan Agricultural University, Chengdu 611100, China; 2Sichuan SEP Analytical Services Co., Ltd., Chengdu 610000, China

**Keywords:** tetracycline, metronidazole, *Chlorella pyrenoidosa*, single toxicity, combined toxicity

## Abstract

Antibiotics have become a new kind of organic pollutant as they are widely used in the water environment of China. Tetracycline (TC) is a class of broad-spectrum antibiotics produced or semi-synthesized by actinomycetes. Metronidazole (MTZ) is the first generation of typical nitroimidazoles. The content of nitroimidazoles is relatively high in medical wastewater, and their ecotoxicity is worthy of attention because they are difficult to completely eliminate. In this paper, the effects of TC and MTZ on the growth, cell morphology, extracellular polymer and oxidative stress of *Chlorella pyrenoidosa* (*C. pyrenoidosa*) were studied, and the toxic interactions between TC and MTZ mixture components were analyzed. The results showed that the 96h-EC_50_ of TC and MTZ was 8.72 mg/L and 45.125 mg/L, respectively. The toxicity of TC to *C. pyrenoidosa* was higher than that of MTZ, and the combined toxicity effect of TC and MTZ was synergistic after the combined action of a 1:1 toxicity ratio. In addition, the algal cells of *C. pyrenoidosa* died to varying degrees, the membrane permeability of algal cells was increased, the membrane was damaged, the surface of algal cells exposed to higher concentration of pollutants was wrinkled, and their morphology was changed. The extracellular polymer of *C. pyrenoidosa* was affected by a change in concentration. The effect of pollutants on the reactive oxygen species (ROS) level and malondialdehyde (MDA) content of *C. pyrenoidosa* also had an obvious dose–effect relationship. This study contributes to the assessment of the possible ecological risks to green algae due to the presence of TC and MTZ in aquatic environments.

## 1. Introduction

Antibiotics are organic substances produced by living organisms in their life activities, which can selectively inhibit or affect other biological functions at low concentrations [1]; they are widely used in human daily life. Antibiotics are widely used, difficult to degrade, and have long environmental persistence. When they are discharged into the water environment, they will have a profound impact on aquatic organisms, leading to changes in the structure and function of the biological community in the water environment.

Tetracycline (TC) is a kind of broad-spectrum antibiotic produced or semi-synthesized by actinomycetes. The production, sale and usage of TC is quite extensive in China [2]. Relevant studies have shown that, in some cities in China that use reclaimed water to recharge groundwater, the highest concentration of TC in reclaimed water is 39 ng/L, corresponding to 48 ng/L TC concentration in groundwater [3]. The concentration of TC detected in livestock wastewater in Shanghai was also higher than 100 μg/L [4]. TC has serious toxic effects on humans and animals [5], and fish living in TC-polluted water suffer neurotoxic effects [6]. Some studies have shown that TC can induce the increase of reactive oxygen species (ROS) at the concentration of 20 μg/L, thereby affecting the development of pufferfish and larvae [7]. TC can also induce an oxidative stress response by changing the activities of superoxide dismutase (SOD) and catalase (CAT) in fish and other organisms [6]. In the clinical treatment of periodontal disease, TC and metronidazole (MTZ) are two commonly used drugs, so the content of both in the medical treatment water is relatively high. MTZ belongs to the nitroimidazole class of drugs, which have been found to have carcinogenic, mutagenic and toxic properties; nevertheless, MTZ is still widely used, especially in veterinary practice and aquaculture, to improve survival rates [8,9]. A large part of the MTZ consumed by humans and animals is not metabolized by the body, and the residue can accumulate in the aquatic environment after entering the water environment through excretion [10,11], which has a significant impact on aquatic organisms. Troc et al. [12] investigated and found that the metronidazole concentration in water before and after treatment at Spanish sewage plants was 164 ng/L and 167 ng/L, respectively. Through the detection of the effluent from the sewage treatment plant, it was found that the concentration of MTZ had reached a level that could cause harmful effects towards humans and the ecological environment [13].

*Chlorella pyrenoidosa* (*C. pyrenoidosa*) belongs to the phylum Chlorophyta and is widely distributed and has strong reproductive ability. It is the main primary producer and plays an important role in the balance and stability of aquatic ecosystems. When a large amount of organic matter enters the water, it will affect the growth of algae. Therefore, the degree of water pollution can be predicted by the change of algae in the water [6]. *C. pyrenoidosa* is often used in aquatic ecology toxicity experiments because it can be easily cultured in the laboratory and can play a role of purification and indication in water self-purification.

Up to now, most studies have focused on the effects of TC on algae, but studies on the effects of MTZ on algae are relatively few. A large number of studies have found that when TC and MTZ are released into water in large quantities, they will accumulate in algal cells and have a negative impact on aquatic ecosystems. Therefore, *C. pyrenoidosa* was selected as the research object in this study, and a classical algal growth inhibition experiment was used to evaluate the single and combined toxic effects of TC and MTZ on *C. pyrenoidosa* combined with changes in the biochemical indicators of algal cells. The number of algal cells, cell morphology, extracellular polymers, ROS levels, malondialdehyde (MDA) content and cell membrane permeability of *C. pyrenoidosa* were determined, and the combined action type of TC and MTZ was evaluated. This study provides insights for the comprehensive risk assessment of TC and MTZ in aquatic ecosystems.

## 2. Materials and Methods

### 2.1. Drug and Microalgae Culture

TC and MTZ: purity > 99.0%, purchased from Shanghai Aladdin Biochemical Technology Co., LTD. (Shanghai, China) *C. pyrenoidosa* (FACHB-27) was selected for this experiment and was purchased from Institute of Hydrobiology, Chinese Academy of Sciences. After BG11 conversion, it was placed in a biochemical incubator with light/dark (12 h:12 h), temperature 26 °C and light intensity 1900–2200 Lx. The density of algal cells was observed under a microscope, and the absorbance value at 680 nm was measured using a UV spectrophotometer (model 752, China). The linear regression equation between OD_680_ value and cell density was as follows: cell density (×10^6^ cells/mL) = 28.05 × OD_680_ + 0.265 (R^2^ = 99.8%), and the OD_680_ of algal liquid in the experiment was 0.25.

### 2.2. Algal Cell Density and Median Effective Concentration (EC_50_)

The test period was 7 days. A certain amount of experimental algal liquid was taken, and the algal cell density of the experimental samples was obtained using the standard curve of algal density (Microsoft Corporation, Redmond, WA, USA).

In the EC_50_ experiment for TC, nine concentration gradients were set: 0 mg/L, 1 mg/L, 2 mg/L, 5 mg/L, 10 mg/L, 15 mg/L, 20 mg/L, 25 mg/L and 30 mg/L. In the EC_50_ experiment for MTZ, 9 concentration gradients were set: 0 mg/L, 10 mg/L, 20 mg/L, 30 mg/L, 40 mg/L, 50 mg/L, 60 mg/L, 70 mg/L and 80 mg/L. At 96 h, samples were taken, OD_680_ was measured, the density of algal cells was calculated, and the inhibition rate of growth compared with the blank control group was calculated. The logarithm value of pollutants was the abscissa, and the inhibition rate of growth was the ordinate. The 96h-EC_50_ of TC and MTZ was obtained by Origin software fitting. The 96h-EC_50_ of TC combined with MTZ was obtained using the same method.

### 2.3. Combined Toxicity Effect Evaluation

In this experiment, after obtaining the 96 h median effective concentration (EC_50_) of *C. pyrenoidosa* under the single and combined toxic effects of TC and MTZ, the combined toxic effects of TC and were evaluated by combining the toxicity unit method (TU) and the additive index method (AI) [14]:$${\mathrm{TU}}_{i}=\frac{C_{i}}{{\mathrm{EC}}_{50i}}$$
$${\mathrm{TU}=\Sigma \mathrm{TU}}_{i}$$
M=TU
where C_i_ is the concentration of compound I, and EC_i_ is the EC_50_ of compound I.

Evaluation criteria of combined effects of additive index AI on poisons:$$M\;\leq \;1:\;\mathrm{AI}=\left(\frac{1}{M}\right)-\;1.0$$
(1)M>1: AI=1.0+(−1) M

When AI = 0, AI > 0 and AI < 0, the combined toxic effects are simple summation, synergism and antagonism, respectively.

### 2.4. Toxicity Test

#### 2.4.1. Single Toxicity Assay of TC and MTZ

The experimental period was 7 days, and 1 blank group and 5 experimental groups were set. *C. pyrenoidosa* in the logarithmic growth period was used to make the initial density of algae inoculated by the algae species 7.0 × 10^5^ cells/mL, and then, the dissolved TC was added to the algae liquid to make the concentration of TC reach 1.0, 2.0, 5.0, 10.0 and 15.0 mg/L, respectively. Algae without TC (0 mg/L) were used as a blank control group and exposed to microalgae culture conditions for 7 days, and each experimental concentration was repeated three times.

The concentration of MTZ was set to 10.0, 20.0, 30.0, 40.0 and 50.0 mg/L, respectively. The other specific procedures were the same as for the TC single toxicity exposure experiment.

#### 2.4.2. Combined Toxicity Experiment

The single-compound 96h-EC_50_ of TC and MTZ on *C. pyrenoidosa* was each defined as one toxicity unit (1TU) and was combined according to the toxicity unit ratio of 1:1. Five concentration combinations were set within the toxicity ratio, and the concentration combination was defined as T. They were defined as 0 T, 0.04 T, 0.4 T, 1 T, 2 T and 3 T, respectively, according to the concentration gradient (Table 1). The combined toxicity effect was evaluated according to the addition index (AI) method [15] of aquatic toxicology. For testing the toxic effects of TC and MTZ on *C. pyrenoidosa* in the algal liquid culture system, the concentration of the mixed system was made to reach 0.04 T, 0.4 T, 1 T, 2 T and 3 T, respectively. The algae without TC and MTZ were used as the blank control group (0 T). The other specific operations were the same as for the TC single toxicity exposure experiment.

### 2.5. Characterization of Algal Cells and Determination of Extracellular Polymers

The effects of pollutants on the surface morphology and extracellular polymers of algal cells were investigated using scanning electron microscopy and fluorescence spectrophotometer. After centrifugation of the algal liquid on day 7 of the experiment at 4000 rpm for 10 min, the supernatant was discarded, the pellet was resuspended in 0.05 M phosphate-buffered saline (PBS), and the surface characteristics of algal cells were observed by the method of Mao [16]. After centrifugation of the algal cell suspension at 4000 rpm for 10 min, the supernatant was supplied, and the filtrate was shaken well for three-dimensional fluorescence spectroscopy.

### 2.6. Oxidative Stress Analysis

The degree of damage caused by the pollutants to the antioxidant system of algal cells was studied by measuring the changes in reactive oxygen species and MDA content. 2′,7′-Dichlorodihydrofluorescein diacetic acid (DCFH-DA) was added to the algal fluid, and the ROS level of *C. pyrenoidosa* algal cells was measured by microplate reading at the maximum excitation wavelength of 480 nm and the maximum emission wavelength of 525 nm [17]. An appropriate amount of algal liquid was taken and centrifuged at 4500 rpm for 10 min, and the supernatant was poured out. The supernatant was suspended and washed twice with 0.1 mol/L PBS with pH 7.4, and then centrifuged to collect the algal cells. After the liquid nitrogen volatilized, an appropriate amount of PBS was manually homogenized for 15 min, and then centrifuged at 2500 rpm for 10 min. The supernatant was taken for the determination of MDA content with a kit (Nanjing Jiancheng Bioengineering Institute).

### 2.7. Determination of Cell Membrane Permeability

In order to investigate the effect of TC and MTZ toxicity on the integrity of the *C. pyrenoidosa* cell membrane, the permeability of the cell membrane was measured. After centrifugation at 4000 rpm for 10 min, the supernatant was poured out, fluorescein diacetic acid (FDA) was used for fluorescence labeling, and the membrane permeability of *C. pyrenoidosa* was measured by microplate reading according to the method of Cai [18].

### 2.8. Data Analysis Method

Three parallel experiments were set for all the above experiments, and the experimental results were the average of the three experiments. Data were analyzed by one-way analysis of variance using SPSS 17.0. Different letters or *p* < 0.05 indicated significant differences between the results. Origin 2018 was used to plot the data, where the error bars represent the standard deviation (SD) between three experiments.

## 3. Results and Discussion

### 3.1. Influence on the Growth of C. pyrenoidosa

As seen from Figure 1A, different experimental time and different TC concentration showed a certain “dose–time–effect” relationship on *C. pyrenoidosa*. On day 2, there was no significant difference in algal cell biomass in the 1 mg/L treatment group compared with the control group (*p* > 0.05). On the fourth day, the effect of 1 mg/L treatment on algal cell biomass changed from a promoting effect to an inhibiting effect due to prolonged action time, and the difference was significant (*p* < 0.05). On the seventh day, compared with the control group, the inhibition rates of *C. pyrenoidosa* cells in the 5 mg/L, 10 mg/L and 15 mg/L treatment groups all reached the maximum, which were 63%, 76% and 84%, respectively, which may be due to the ability of antibiotics to damage the cell wall and membrane structure [19,20]. With the increase of TC concentration, the damage to the cell wall and membrane of *C. pyrenoidosa* became greater. According to Figure 1B, the influence of MTZ on the growth of *C. pyrenoidosa* conforms to the law of “low promotion and high inhibition”. On the second day, the 10 mg/L and 20 mg/L treatment groups were 6.6% and 0.8% higher than the blank group, respectively, which showed a promoting effect on the growth of algal cells, while the other treatment groups showed an inhibiting effect, indicating that lower concentration of MTZ stimulated algal cell division and, finally, showed a promoting effect. As time went on, the inhibition rates of the 20 mg/L, 30 mg/L, 40 mg/L and 50 mg/L treatment groups all reached their peaks on the seventh day, which were 25%, 51%, 68% and 85%, respectively. This may be because the high concentration of MTZ inhibited the synthesis of chlorophyll A and caused different degrees of damage to the photosynthetic system of the algal cells, which, in turn, leads to a decrease in algal cell density [21].

According to Figure 1C, when TC and MTZ were combined, the algal cell density of the 0.04TU treatment group was 7% higher than that of the control group on the first day. On the second and third day, 0.04 T treatment group continued to promote the growth of algal cells with promotion rates of 4% and 1%, respectively. This may be because in the combined action, the combination of the two pollutants causes irreversible internal damage to the *C. pyrenoidosa* cells; the algae cells cannot repair this damage by their own ability, so the promotion phenomenon gradually weakened. Different from TC and MTZ single action, on the seventh day, the inhibition rate continued to increase in all treatment groups, the reason may be that when combined, the toxicity of the two pollutants is very strong and will not change because of other repair capacity of algal cells, so the inhibition rate of *C. pyrenoidosa* will only increase with time.

### 3.2. Evaluation of the Combined Toxic Effect of TC and MTZ on C. pyrenoidosa

One of the most commonly used parameters in toxicity determination is the median effective concentration (EC_50_). The 96h-EC_50_ of *C. pyrenoidosa* under MTZ and TC single and combined is shown in Figure 2A–C. The 96h-EC_50_ of TC and MTZ was 8.720 mg/L and 45.125 mg/L, respectively (Figure 2A,B). The toxicity of TC is about 5.2 times that of MTZ. The 96h-EC_50_ of *C. pyrenoidosa* under combined stress of the two pollutants was 1.066 T (Figure 2C). The combined toxicity evaluation method was used to evaluate the combined toxicity effect. The results showed that when the mixed toxicity ratio of TC and MTZ was 1:1, the combined toxicity effect of TC and MTZ on *C. pyrenoidosa* showed a synergistic effect, and the order of toxicity was as follows: combined toxicity > single toxicity, TC > MTZ.

### 3.3. Effect on Morphology of C. pyrenoidosa

According to the SEM results, in the blank group (Figure 3D), *C. pyrenoidosa* cells showed an ellipsoid shape, and the surface of the cells was relatively smooth; the cell structure was complete, and no other particles were obviously observed. In the TC treatment group (Figure 3A), *C. pyrenoidosa* was prone to agglomeration; severe folds appeared on the surface of the algal cells, cell rupture could be clearly observed and the cell surface could not maintain integrity. This may be due to the physical damage caused by TC to the algal cells, which are irreversibly destroyed [22], and because tetracycline antibiotics inhibit the growth of bacteria by preventing the synthesis of proteins [23], thus destroying the morphological structure of algal cells. As shown in Figure 3B, the MTZ treatment group could also cause severe folds on the cell surface of *C. pyrenoidosa*, and the algal cells also appeared to agglomerate; this may be because the cell damage caused by MTZ to *C. pyrenoidosa* resulted in membrane deformation and damage and cell-content leakage [24] and, finally, the deformation of algal cells. However, compared with the influence of TC mentioned above, the aggregation of algal cells was not obvious, the surface folds were not serious, and there was no rupture phenomenon, which also indicated that MTZ had a lesser influence on *C. pyrenoidosa*, and its toxicity was weaker than that of TC.

Figure 3C illustrates the effect of TC and MTZ on the morphology of *C. pyrenoidosa*. Due to the strong toxicity under the combined effect of TC and MTZ, most of the *C. pyrenoidosa* algal cells were broken, and the algal cells dried up, with a larger pore size on the surface. The morphological and structural changes caused by TC and MTZ were more serious than those caused by TC and MTZ single action, which indicated that the toxicity of TC and MTZ to *C. pyrenoidosa* would be enhanced under their combined action, which is consistent with the conclusion obtained in Section 3.2. The combined effect of TC and MTZ causes a certain degree of damage to the cell walls and membranes of microalgae, which may be caused by the strong oxidative stress response induced by their combined effect or the external expression of microalgae to cope with the combined toxicity [25].

### 3.4. Effect on the Extracellular Polymer of C. pyrenoidosa

In this experiment, three-dimensional fluorescence technology was used to analyze the changes of EPS secreted by *C. pyrenoidosa* under the effect of TC (Figure 4A). The peak strength of the TC group processing in the area I, II and IV was lower than the control group (Figure 4D). This shows that TC inhibits the synthesis of proteins and soluble microbial products, at the same time, the area V peak intensity is abated, causing humic acid organic matter to almost disappear. A study [26] found that humic acid organic compounds can inhibit the growth of algae; however, since the organic matter disappeared in our experiment. it can be speculated that high concentration TC no longer inhibited growth by inducing algal cells to synthesize humic acid due to its strong toxicity. There may be various ways to inhibit algal cell growth, e.g., by affecting cell membrane permeability. The peak intensity in area IV of the MTZ treatment group (Figure 4B) was lower than that of the control group, which inhibited the synthesis of dissolved microbial products of *C. pyrenoidosa*. At the same time, the peak intensity of area V was enhanced, which indicated that MTZ could induce the synthesis of humic acid organic matter, thus inhibiting the growth of algal cells.

The peak intensity of the mixed system in area I and II was low, and the peak almost disappeared (Figure 4C), indicating that the synthesis of *C. pyrenoidosa* was seriously blocked under the combined action of TC and MTZ, which seriously hindered the synthesis of soluble microbial products. It indicates that after mixing TC and MTZ, the growth environment of *C. pyrenoidosa* was seriously affected, and the algal cells could no longer synthesize lytic microorganisms to regulate the effects of the environment [27]. Compared with the control group, the peak intensity in area III and V was enhanced, which may be due to the synthesis of fulvic acid and humic acid organic matter by algal cells affected by TC and MTZ and the inhibition of the growth of algal cells. This is also consistent with the experimental conclusion shown in Section 3.1 that the 3 T treatment group seriously affected the growth of algal cells.

### 3.5. Effects on the Antioxidant System of C. pyrenoidosa

TC and MTZ are two different antibiotics, the sensitivity of *C. pyrenoidosa* to different antibiotics is quite different, and this difference may be caused by the oxidative stress on algal cells induced by foreign pressure.

As seen from the experimental results in Figure 5A, on the seventh day of the experiment, the ROS content of the TC treatment group for all concentrations was always higher than that of the control group, and there was a significant difference (*p* < 0.05). Therefore, it can be concluded that the ROS content in *C. pyrenoidosa* cells was stimulated by TC treatment at all concentrations. Among them, the 10 mg/L and 15 mg/L treatment groups had the highest ROS content, which was 4.3 and 4.6 times that of the control group, respectively. This indicates that a high concentration of TC has a greater impact on the environment of algal cells, which may also cause the algal cells to produce more H_2_O_2_, O^2^·O^2−^ and ·OH [28]. When the intracellular ROS level is too high, it will react with cell membrane tissue or organelle membrane lipids to form lipid peroxidation products (MDA). The experimental results showed that TC could affect the MDA content of *C. pyrenoidosa*. The MDA content of the 10 mg/L and 15 mg/L treatment groups was significantly higher, by 3.7 and 5.4 times, respectively. The generated MDA can interact with other biological macromolecules, e.g., cross-linking with proteins to make them lose their function, which seriously interferes with the normal physiological and metabolic processes in cells. There is an obvious “dose–effect” relationship between MTZ and ROS content (Figure 5B). However, different from the influence caused by TC, the ROS content did not increase suddenly due to an increase in the concentration of MTZ, which may be because the toxicity of MTZ is less than that of TC. When the concentration reached 40 mg/L, the ROS content was three times that of the control group. MTZ rapidly increases the ROS production in *C. pyrenoidosa* and accumulates in the algae cells to a certain extent, which will lead to the breaking of the body balance, thereby attacking biomolecules and causing damage to the body. The amount of ROS produced is so excessive that the antioxidants produced by the cell are not enough to remove it, and oxidative stress reaction will occur in the cell body [29]. Therefore, when MTZ enters the body of *C. pyrenoidosa*, it will have a serious impact on the algal cells. The MDA content of MTZ reached the maximum at 50 mg/L, which was 5.1 times that of the control group. The increase of MDA in algal cells increases the level of lipid peroxides in the cells, which may cause cell membrane damage, thus inhibiting the normal growth and metabolic activities of *C. pyrenoidosa*.

Compared with the control group, when the concentration of the mixed system was higher (Figure 5C), which can cause the algal intracellular ROS levels to rise, begin to withstand the antioxidant enzyme system, cause cell mitochondria damage, lipid peroxidation, and so on, so that the cell function is lost, until it causes apoptosis [30]. Therefore, the ROS content of the 2 T and 3 T treatment groups was significantly increased, which was 3.7 and 4.9 times higher than that of the control group. The change trend of MDA content was consistent with that of the ROS content. Compared with the control group, the MDA content of the 3 T treatment group reached 5.7 times, which was more significant than the effect of TC and MTZ alone, indicating that the ROS content increased by the mixed TC and MTZ beyond the range that algal cells could withstand and aggravated the damage due to membrane lipid peroxidation. The normal physiological and metabolic levels of algal cells collapsed, and *C. pyrenoidosa* was severely damaged.

### 3.6. Effect on Cell Membrane Permeability of C. pyrenoidosa

If the wall of an algal cell is the first protective barrier, the cell membrane is the second. When pollutants come into contact with algal cells, they may destroy part of the cell wall and contact the cell membrane, resulting in cell membrane damage and even loss of membrane integrity. As seen from Figure 6A, compared with the control group, on the first day, the low concentration of TC (1 mg/L) did not damage the cell membrane of *C. pyrenoidosa*. With the extension of time, the influence of TC on the membrane permeability of *C. pyrenoidosa* had a “dose–effect” relationship. On the seventh day, with the increase of time and concentration, ROS and other substances in algal cells attacked the organelle membrane and the cytoplasmic membrane, resulting in a lipid peroxidation reaction, and the normal function of the cell membrane was destroyed. Therefore, the effect of the 15 mg/L treatment group on cell membrane permeability reached the maximum, 5.4 times that of the control group, and the cell membrane was seriously damaged, which also reflected the basic death of algal cells at this concentration [31]. On the seventh day, the effect of 10 mg/L MTZ on cell membrane permeability became slower (Figure 6B), which may be because *C. pyrenoidosa* gradually adapted to the environment of low-concentration MTZ with the prolongation of time, and the damage due to low-concentration MTZ to algal cells was limited. Therefore, with the change of time, the effect of the treatment group on the membrane of algal cells was not obvious. The cell membrane permeability of the 50 mg/L treatment group was 2.5 times that of the blank control group. Compared with the effect of TC, the effect of MTZ was smaller, which may be due to the greater toxicity of TC than MTZ and the greater influence of TC on the cell membrane permeability of *C. pyrenoidosa*.

The combined effect on cell membrane permeability of *C. pyrenoidosa* was different from that of single effect (Figure 6C). On the first day, the 0.04 T and 0.4 T treatment groups had little effect on cell membrane permeability with change of time, which may be because the low-concentration treatment group caused *C. pyrenoidosa* to produce other EPs. These EPs can reduce the permeability of the cell membrane or improve the permeability of the cell membrane appropriately and improve the tolerance of algal cells to the environment [32]. The cell membrane permeability of the 1 T, 2 T and 3 T treatment groups on the seventh day was 2.8, 4.0 and 5.8 times higher than that of the control group, respectively. Compared with the single action, the combined action had a greater effect on cell membrane permeability and caused more serious damage to the cell membrane.

## 4. Conclusions

In this study, we investigated the toxic effects of TC and MTZ on *C. pyrenoidosa,* alone and in combination, and their mechanisms. The results showed that the 96h-EC_50_ of TC and MTZ was 8.72 mg/L and 45.125 mg/L, respectively, which seriously affected the growth and antioxidant mechanism of *C. pyrenoidosa*. When TC and MTZ were mixed, the toxic effects of the two substances on *C. pyrenoidosa* were changed. When the mixed toxicity ratio was 1:1, the type of combined toxicity was synergistic. The results of this experiment are helpful to further understand the toxic effects of TC and MTZ on marine microalgae, and they have important significance for evaluating the toxicity of marine microalgae.

## Figures and Tables

**Figure 1 ijerph-20-03623-f001:**
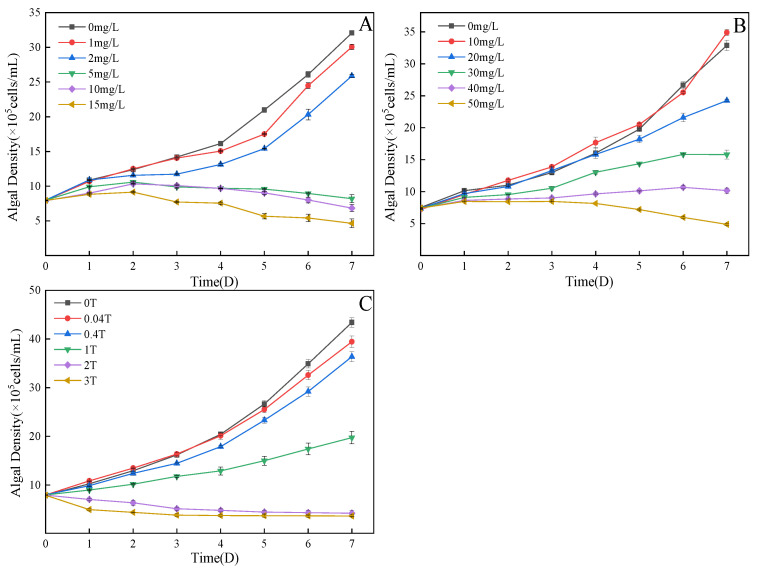
Effects on the growth of *C. pyrenoidosa*: (**A**) TC; (**B**) MTZ; (**C**) TC + MTZ.

**Figure 2 ijerph-20-03623-f002:**
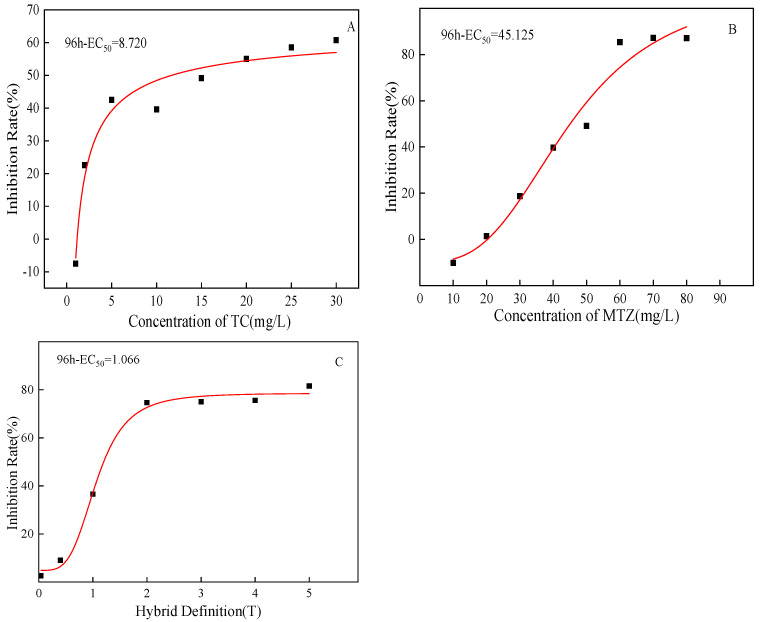
The fitted curve of 96h-EC_50_: (**A**) TC; (**B**) MTZ; (**C**) TC + MTZ.

**Figure 3 ijerph-20-03623-f003:**
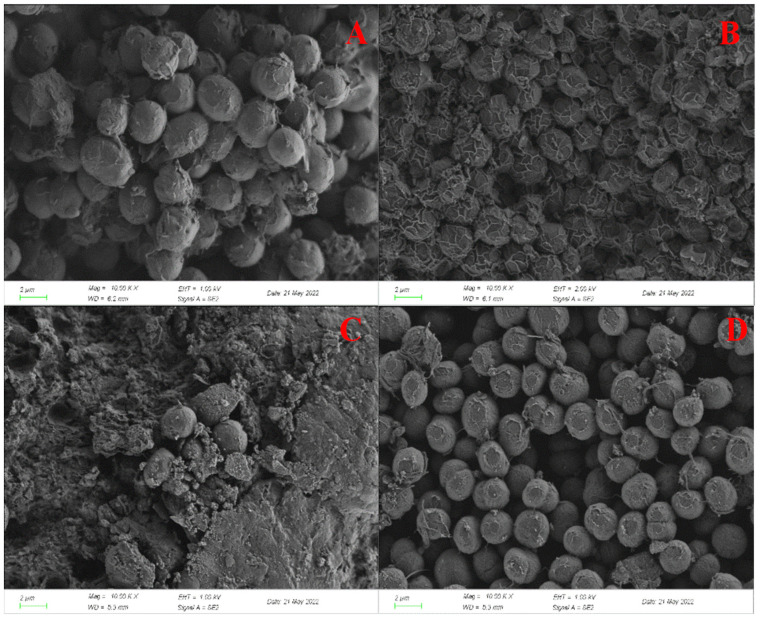
SEM images of *C. pyrenoidosa* exposed for 96 h: (**A**) 15 mg/L TC; (**B**) 50 mg/L MTZ; (**C**) 3 T; (**D**) blank control group.

**Figure 4 ijerph-20-03623-f004:**
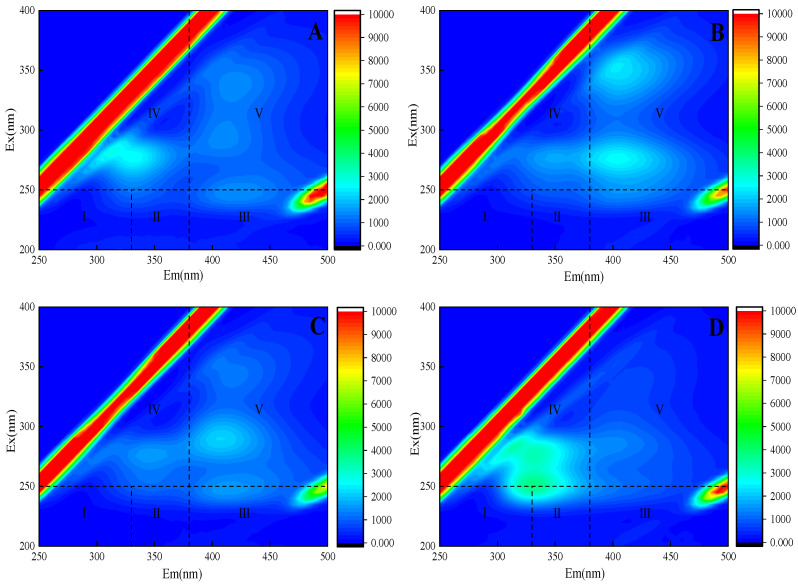
3D-EEM fluorescence spectra of *C. pyrenoidosa* exposed for 96 h: (**A**) 15 mg/L TC; (**B**) 50 mg/L MTZ; (**C**) 3 T; (**D**) blank control group.

**Figure 5 ijerph-20-03623-f005:**
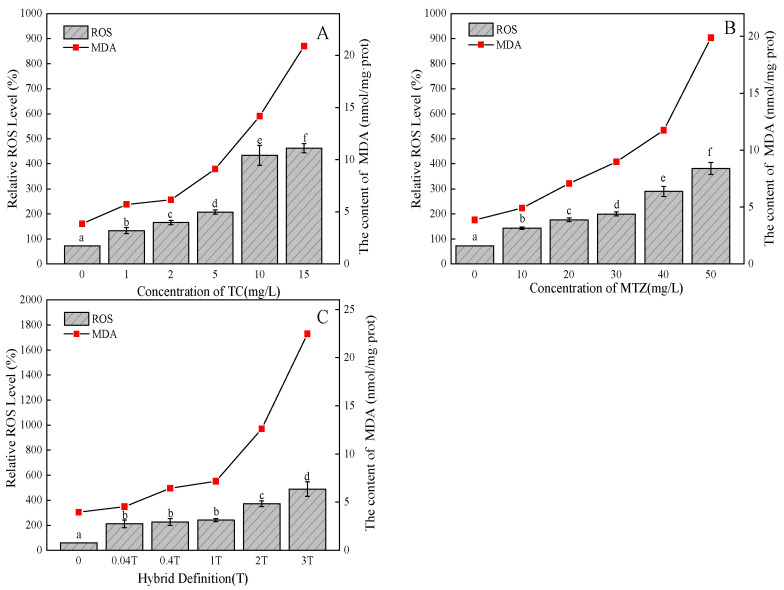
Effects on ROS and MDA contents of *C. pyrenoidosa*: (**A**) TC; (**B**) MTZ; (**C**) TC + MTZ. Note: Data are represented by means + SD. Different lowercase letters indicate significant difference between treatment groups (*p* < 0.05).

**Figure 6 ijerph-20-03623-f006:**
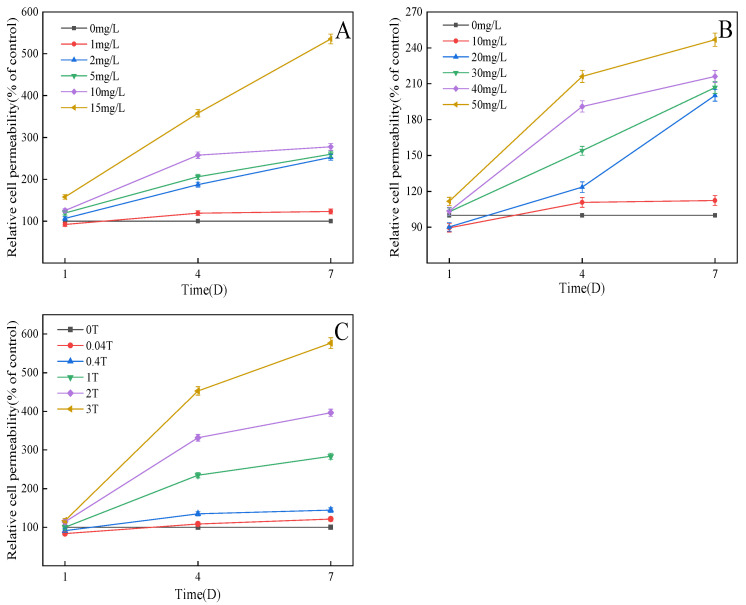
Effects on cell permeability of *C. pyrenoidosa*: (**A**) TC; (**B**) MTZ; (**C**) TC + MTZ.

**Table 1 ijerph-20-03623-t001:** Concentration combination of TC and MTZ at 1:1 toxicity.

Toxicity Ratio (TC:MTZ)		Concentration Combination (mg/L)					
	Toxic	0 T	0.04 T	0.4 T	1 T	2 T	3 T
1:1	TC	0	0.1	1	2.5	5	7.5
	MTZ	0	0.52	5.2	13	26	39

## Data Availability

Not applicable.

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
