# Peer review of "Toxicity of Tetracycline and Metronidazole in Chlorella pyrenoidosa"

_ijerph, 2023, doi:10.3390/ijerph20043623_

Round 1

Reviewer 1 Report

1. The manuscript deals with the problem of the toxicity of tetracycline and metronidazole on the growth and development of the microalgae Chlorella pyrenoidosa.

2. The Materials and Methods section contains a series of mistakes or confusing formulations that the authors should return to, either to correct them or to clarify them:

a. line 86 - the absorbance of the algal culture is read at 680 nm (visible spectrum) with a UV spectrophotometer;

b. line 91 - "the culture period was 7 days". I think it is better to reformulate the sentence "Cultivation of microalgae and observations were made over a period of 7 days". This is because culture is not the same thing as cultivation;

line 99 - incorrect "abscess" (Abscess= swollen area within body tissue, containing an accumulation of pus); correct is abscissa

 line 136 - I would suggest the authors to detail Mao's method;

 line 147 - the same, details regarding the determination of the MDA content;

 line 152 - technical details regarding Cai's method;

 line 157 - the statement " Different letters or P<0.05 indicated significant differences between the results. " needs clarification.

 3. Results and discussions section

lines 183-186 - the statement "growth promotion" is incorrect because in the experimental variants with antibiotics the growth is weaker compared to the control and indicates inhibition.

 The effect of tetracycline is exerted mainly on protein synthesis. That is why it would have been more useful to evaluate the ribosomal activity in this case.

 4. The Conclusions section should be more consistent and include very briefly the results regarding the effect of antibiotics on changes in cellular morphology and the antioxidant system of microalgae.

Reviewer 2 Report

By setting up single and combined contamination experiments, the authors measured and studied the effects of TC and MTZ on C. pyrenoidosa growth, cell morphology, extracellular polymer, oxidative stress response and cell membrane permeability, etc., so as to explore the apparent and toxic effects. However, there are the following questions for the author's reference to answer or modify.

1. There are many problems in the writing format of the full manuscript (part of which is highlighted in yellow). Please revise the full text carefully.

2. The innovation of the manuscript needs to be improved, the overall innovation point could not be refined, and the setting of toxicity research needs to be considered.

3. The depth of research can be further deepened.

4. Line 34-36: Is it possible to separately describe the effects of TC and MTZ on aquatic ecology, the toxicity to aquatic organisms, the pollution levels, and the specific impacts on biological communities, etc., here or in the second paragraph? Please revise it carefully.

5. The study only conducted experiments on C. pyrenoidosa and measured relevant indicators (including oxidative stress response, etc.,), while other toxicity indicators were not considered, which could not be related to the structure and function of the biological community in the aquatic environment. Please consider increasing relevant studies on the food chain or community? And please revise it carefully.

6. It is suggested to add the introduction of TC and MTZ combined pollution in aquatic environment in the second paragraph of the section introduction, so as to prove the necessity and significance of studying combined pollution. There are many kinds of aquatic environment and the range is very wide. How to determine the situation of combined pollution? Please revise it carefully.

7. Line 64-65: It is suggested to add literature support and examples of C. pyrenoidosa used in aquatic ecological experiments.

8. The evaluation of related indicators of a single alga (C. pyrenoidosa) seems to be insufficient to be used as a comprehensive risk assessment of TC and MTZ in aquatic ecosystems. Could the objects of study represent the whole water ecological environment or the biological community structure? Please revise it carefully.

9. Such as cell density, EC50, characterization of algal cells, extracellular polymers, oxidative stress, and cell membrane permeability, the toxic effect of represented by these indicators has not been clearly distinguished and expressed, and the relationship with single toxicity and combined toxicity has not been clearly expressed. The setup and order of research statements is a little confusing. Please revise it carefully.

10. Line 94: Please explain the basis for the different experimental concentration settings of TC and MTZ.

11. Line 101: Please give the software information.

12. The Addition Index (AI) method is not clear enough. Is there any calculation basis or formula?

13. Line 197-198: Please add some literature support.

14. Figure 5: not clear, please change the figure with higher resolution.

Round 2

Reviewer 2 Report

  • The author gave careful answers to the questions.